# The Identification and Role of the Key Mycotoxin of *Pestalotiopsis kenyana* Causing Leaf Spot Disease of *Zanthoxylum schinifolium*

**DOI:** 10.3390/jof9121194

**Published:** 2023-12-13

**Authors:** Chang Liu, Yiling Li, Hang Chen, Shuying Li, Shan Han, Tianhui Zhu, Yinggao Liu, Shujiang Li

**Affiliations:** College of Forestry, Sichuan Agricultural University, Chengdu 611130, China; lc10239428@163.com (C.L.); liyiling1@foxmail.com (Y.L.); 18181648753@163.com (H.C.); leesy0710@126.com (S.L.); hanshan6618@163.com (S.H.); zhuth1227@126.com (T.Z.); lyg927@263.net (Y.L.)

**Keywords:** prickly ash, mycotoxins, pestalopyrone, toxicity

## Abstract

Leaf spot is a common disease of *Zanthoxylum schinifolium* (*Z. schinifolium*), which can seriously harm the plant’s ability to grow, flower, and fruit. Therefore, it is important to identify the mechanism of leaf spot caused by *Pestalotiopsis kenyana* (*P. kenyana*) for thorough comprehension and disease control. In this study, to verify whether the mycotoxins produced by *P. kenyana* cause leaf spot disease, the best medium for *P. kenyana*, namely PDB, was used. The mycotoxins were determined by ammonium sulfate precipitation as non-protein substances. The crude mycotoxin of *P. kenyana* was prepared, and the optimal eluent was eluted with petroleum either/ethyle acetate (3:1, *v*/*v*) and purified by silica gel column chromatography and preparative high-performance liquid chromatography to obtain the pure mycotoxins PK-1, PK-2, and PK-3. The PK-3 had the highest toxicity to *Z. schinifolium*, which may be the primary mycotoxin, according to the biological activity test using the spray method. The physiological and biochemical indexes of *Z. schinifolium* plants treated with PK-3 mycotoxin were determined. Within 35 days after mycotoxin treatment, the results showed that the protein content and malondialdehyde content of leaves increased over time. The soluble sugar and chlorophyll content decreased over time. The superoxide dismutase activity and catalase activity of the leaves increased first and then decreased, and the above changes were the same as those of *Z. schinifolium* inoculated with the spore suspension of the pathogen. Therefore, it is believed that the mycotoxin pestalopyrone could be a virulence factor that helps *P. kenyana* induce the infection of *Z. schinifolium*. In this study, the pathogenic mechanism of *Z. schinifolium* leaf spot was discussed, offering a theoretical foundation for improved disease prevention and control.

## 1. Introduction

Pathogenic fungi produce various compounds that contribute to plant diseases, such as enzymes, hormones, and mycotoxins [1]. Among these, mycotoxins have been extensively studied and are closely associated with disease progression. Mycotoxins are secondary metabolites produced by fungi, and play a crucial role in the development of plant diseases through mutual recognition and interaction between fungi and plants [2]. The majority of mycotoxins are secondary metabolites with low molecular mass that typically consist of peptides or proteins, sugars, lipids, aromatic rings, heterocycles, and organic acids. Numerous common pathogenic fungi produce mycotoxins, such as *Fusarium*, *Cercospora*, *Pestalotiopsis*, *Sclerotinia*, *Alternaria*, *Bipolaris*, *Phoma*, etc. [3,4].

*Pestalotiopsis kenyana* (*P. kenyana*) was identified as the pathogen of *Zanthoxylum schinifolium* (*Z. schinifolium*) leaf spot in our previous study [5]. *P. kenyana* can also damage *Myrica rubra*, *Photinia serrulata*, *Camellia oleifera*, and other plants [6]. *Pestalotiopsis* is a diverse genus of asexual fungi widely distributed in tropical and temperate regions [7]. It produces conidia and is a common plant pathogen causing various diseases, including canker, shoot blight, leaf spot, gray spot, leaf blight, and fruit rot [8,9,10,11,12,13]. *P. sensulato* was identified as the pathogen causing the gray blight of tea in Japan [14]. Additionally, P. theae has been linked to leaf spots in Ixora chinensis, and *Elaeis guineensis* [15,16]. The causal agent of the Black Spot Blight of *Pinus sylvestris* var. mongolica is *P. neglecta* [17].

*Pestalotiopsis* can produce abundant metabolites and can also produce mycotoxins as virulence factors to induce disease symptoms during the infection of host plants. The reported mycotoxins are often ketones [18,19]. Two mycotoxins, oxysporone and pestalopyrone, were isolated from the culture medium of the pathogenic fungus *P. oenotherae* [20]. Oxysporone was also isolated from the pathogenic fungus *P. ongiseta* [21]. *P. guepinii* can produce pestalopyrone to damage hazelnut branches [22]. The pathogenic fungus of tea gray blight, *P. longiseta,* can produce two mycotoxins, oxysporone and (+)-epiepoxydon [23]. In addition, *P. microspora* and *P. funereacan* can also produce mycotoxins and infect plants. However, it has not been figured out which secondary metabolites of *P. kenyana* are the main factors leading to the occurrence of *Z. schinifolium* leaf spot.

During the cultivation of pathogenic fungi, it is common for them to secrete mycotoxins into the surrounding medium [24,25]. For instance, *P. foedan* was found to secrete a monoterpene lactone (1R, 4R, 5R, 8S)-8-hydroxy-4,8-dimethyl-2-oxabicyclo [3.3.1]nonan-3-one in its liquid culture [26]. Additionally, three new dimeric terpenoids, known as Pestalofones I-K, were isolated from the solid culture of *P. fici* [27]. The production of mycotoxins by these fungi is dependent on various factors, such as the type of medium, temperature, time, light, pH, carbon source, nitrogen source, and other external conditions. Among these factors, the type of medium has the greatest influence on mycotoxin production [28,29]. It is worth noting that different fungi require different cultural conditions. The cultural conditions of *P. vismiae* were investigated. The results revealed that the optimal temperature for its growth was 25 °C, while the preferred carbon source was sucrose, and yeast extract was the preferred nitrogen source. The fungus also exhibited the best growth at an initial pH of 6.5 [30]. As for *P. grandiflorum,* it was observed that it grew its mycelium faster in the dark compared to when it was exposed to light [31]. *P. uvicola* demonstrated the ability to produce a neuroprotective terpene trilactone called bilobalide, when cultured in a potato glucose liquid medium [32]. When the pH of the potato dextrose agar (PDA) medium was set to 5.0, *P. mangiferae* exhibited optimal mycelial growth and spore production [33]. Furthermore, *P. mangiferae* was found to produce the highest amount of metabolic extracts under the culture conditions of pH 4.0 and 30 °C [34]. Based on these findings, the aim of this experiment was to screen the best medium for mycotoxin production by culturing the pathogenic fungus in four different media.

To elucidate the role and chemical structure of individual components in the complex mixture of crude mycotoxins obtained from fungal cultures, it is crucial to employ effective methods of separation and purification. Given that mycotoxins are organic substances, conventional organic chemistry and biochemistry techniques can be applied for extraction, separation, and purposes [35]. In one study, *Pestalotiopsis* sp. EJC07 yielded eight compounds from an ethyl acetate extract. These compounds were isolated through gradient elution of a silica gel column chromatography using n-hexane, ethyl acetate, and methanol as eluents [36]. Likewise, the culture medium of *Pestalotiopsis* sp. Z233 was subjected to extraction with ethyl acetate and n-butanol, followed by elution on a Sephadex LH-20 column to obtain two pure compounds [37]. In another experiment, the incubation products were extracted with ethyl acetate and then eluted using a gradient mode of petroleum ether, CH_2_Cl_2_, and MeOH. Subsequently, the fractions were isolated using Sephadex LH-20 column chromatography (CC) and reversed-phase high-performance liquid chromatography (RP-HPLC), leading to the isolation of ten different compounds [38].

Studies have revealed that pathogen mycotoxins produced by pathogens can trigger the generation of reactive oxygen species (ROS) in infected plants. In response, complex physiological and biochemical reactions occur within plant cells to counteract the detrimental effects of ROS. These reactions result in changes in various indicators, including defensive enzymes like superoxide dismutase (SOD), catalase (CAT), peroxidase (POD), and phenylalanine ammonia-lyase (PAL), as well as non-enzymatic substances such as chlorophyll, malondialdehyde (MDA), and total phenols, among others. Through the analysis of these physiological and biochemical indicators, the incidence of plant diseases can be determined, and a better understanding of the interaction mechanism between pathogen mycotoxins and plants can be achieved. By examining the alterations in these indicators following the onset of disease, researchers can gain insights into the complex responses and adaptations that occur within plants as they defend themselves against mycotoxin-induced oxidative stress.

The main objective of this study was to investigate whether *P. kenyana* can produce mycotoxins that would help it cause *Z. schinifolium* leaf spot disease. This study aimed to isolate and purify the mycotoxin produced by *P. kenyana*. To achieve this goal, several steps were followed. Initially, the pathogenicity of *P. kenyana* cultured in different media was assessed, leading to the identification of the most suitable medium. Subsequently, the non-protein nature of the pathogenic components of *P. kenyana* mycotoxin was determined through ammonium sulfate precipitation. The crude mycotoxins were prepared and further purified through TLC, silica gel column chromatography, and HPLC separation. The purified mycotoxins were analyzed to determine their structure. In the final phase of this study, the toxicity of the mycotoxins and their impact on various physiological and biochemical indicators were assessed. This analysis aimed to establish whether the mycotoxins serve as virulence factors that facilitate infection by *P. kenyana*. These findings provide a theoretical basis for the prevention and control of *Z. schinifolium* leaf spot disease of *Z. schinifolium*.

## 2. Materials and Methods

### 2.1. Materials

Plant samples: One-year-old *Z. schinifolium* plants were obtained from Meishan *Z. schinifolium* nursery located in Sichuan, China (30°04′ N, 103°50′ E). These plants, with a height of approximately 30–40 cm, were cultivated in a greenhouse at the Chengdu campus of Sichuan Agricultural University (30°42′ N, 103°51′ E). Within the greenhouse, the temperature was 28 °C, with a relative humidity of 55%.

Fungal culture: *P. kenyana*, isolated from diseased *Z. schinifolium* plants, was provided by the Forest Protection Laboratory of Sichuan Agricultural University [5]. (GenBank accession, ITS: NR147549.1, LSU: MH870724.1, PRB2: MH554958.1, TUB: KM199395.1, and TEF:KM199502.1).

Medium formulas: This study utilized the following medium formulas: (a) Potato dextrose broth (PDB): 200 g potato, 20 g glucose, and filled with distilled water to reach a final volume of 1000 mL. (b) Potato sucrose broth (PSB): 200 g potato, 20 g sucrose, and filled with distilled water to reach a final volume of 1000 mL. (c) Modified martin broth: 20 g glucose, 5 g peptone, 2 g yeast extract powder, 1 g potassium dihydrogen phosphate, 0.5 g magnesium sulfate, and filled with distilled water to reach a final volume of 1000 mL, adjusted to pH 6.4. (d) Czapek dox broth: 2 g sodium nitrate, 1 g dipotassium hydrogen phosphate, 0.5 g potassium chloride, 0.5 g magnesium sulfate, 0.01 g iron sulfate, 30 g sucrose, and filled with distilled water to reach a final volume of 1000 mL, adjusted to pH 7.0.

### 2.2. Screening of the Best Medium for P. kenyana to Produce Mycotoxin and Determination of the Pathogenic Components of the Mycotoxin

#### 2.2.1. Preparation of Mycotoxin Stock Solution and Determination of Toxicity

After activation, pathogen strains were transferred to potato glucose agar medium and cultured at 25 °C for 5 days. Once the colonies were fully grown, a 6 mm diameter piece of fungus was punched out and inoculated into various broths (potato dextrose, potato sucrose, modified martin, and czapek dox). The broths were incubated at 28 °C and 150 r/min for 18 days, with one fungus piece added to each 100 mL of broth. The fungus metabolites were obtained by filtering the mycelium with double-layer gauze, resulting in a concentration of 50 μg/mL (mycotoxin stock solution). In total, 100 mL of the mycotoxin stock solution was taken from each medium for subsequent toxicity testing [39]. To test toxicity, 50 leaves of *Z. schinifolium* with uniform growth and free from pests and diseases were selected. These leaves were sterilized using 75% alcohol and washed three times with sterile water. Each leaf was then sprayed with 1 mL of the mycotoxin stock solution, while sterile water was used as the control. Leaves in vitro infections were moisturized at 25 °C and repeated 10 times [40]. The incidence of leaf lesions was observed, and the percentage of the lesion area to the total leaf area was recorded. The area was calculated using a grid method, where images of the *Z. schinifolium* leaves with leaf spot disease were scanned and printed on grid paper with a known area of 0.04 cm^2^/grid. In order to perform statistical analysis, the full grid was considered as one unit, and more than half but less than one grid was considered as half unit. However, if the lesion covered less than half of a grid, that area was not included in the statistics.

#### 2.2.2. Determination of Toxic Components of Fungal Secondary Metabolites

The ammonium sulfate fractional precipitation method was used in this study. Firstly, 1000 mL of the incubation broth was centrifuged at 4 °C and 4500 r/min for 15 min to collect the supernatant while discarding the fungal mycelium and incubation residue. To precipitate the proteins, ammonium sulfate was slowly added to the supernatant until it reached 20% saturation. The resulting mixture was refrigerated for 2 h and then centrifuged at 12,000 r/min for 15 min. This step separated the supernatant and the precipitate, which were collected separately. The collected supernatant was divided into two parts. One part was used for bioassay addition, while the other part underwent further ammonium sulfate, to achieve saturation at 30%, 40%, 50%, 60%, 70%, and 80%. Each saturated precipitate was collected, dissolved in phosphate-buffered saline (PBS), and stored at room temperature for subsequent analysis [41].

#### 2.2.3. Determination of Biological Activity

The toxicity test conducted in this experiment followed the same procedure as the previous experiment in Section 2.2.1. Seven solutions of saturated precipitation and supernatants from Section 2.2.2 were applied to the leaves in 10 μL droplets. This process was repeated 10 times, resulting in a total of 140 *Z. schinifolium* leaves being used. The incidence of leaf lesions was observed, and the percentage of the lesion area to the total leaf area was recorded.

### 2.3. Purification of Mycotoxins

#### 2.3.1. Preparation of the Crude Mycotoxin Sample

To prepare the mycotoxin stock solution, the method described in Section 2.2.1 was followed. Firstly, the mycotoxin stock solution was subjected to freezing centrifugation at 4 °C and 4500 r/min for 15 min to remove the mycelia. After this step, an equal volume of methanol was added to the centrifuged mycotoxin stock solution. The mixture was shaken to ensure uniform mixing and then centrifuged at 20 °C and 8000 r/min for 15 min. This centrifugation step allowed for the separation into protein and non-protein fractions. The protein fraction was refrigerated for further use, while the non-protein fraction underwent vacuum evaporation at 45 °C to remove the methanol. Once the methanol was removed, the mycotoxin stock solution was extracted twice with half the volume of ethyl acetate. The ethyl acetate phase was separated from the water phase, and the ethyl acetate phase was combined and concentrated to 2 mL by a rotary evaporator.

#### 2.3.2. Separation and Purification by TLC and Silica Gel Column Chromatography

(a) TLC: The crude mycotoxin solution was uniformly spotted at the bottom of an activated silica gel plate, with a distance of 1 cm. The plate was then placed in a saturated tank containing a layering agent to facilitate separation. Once the developing agent had migrated a distance of 1 cm from the bottom, the plate was removed, and the front position was marked. After air drying, the plate was exposed to ultraviolet light for color development. The distribution of samples on the thin layer plate was observed and photographed, and the Rf value of the main components was calculated using the formula Rf = distance traveled by the solute/distance traveled by the solvent. The best-developing agent was selected based on the optimal Rf value and ΔRf value using the solvent system detailed in Appendix A [42].

(b) Silica gel column chromatography: The crude mycotoxin sample was dissolved in methanol, and an appropriate amount of silica gel was added and dried to ensure the adsorption of the mycotoxins. The sample was then loaded onto a chromatographic column packed with silica gel of 100–200 mesh size. Elution was carried out using a large volume of eluent, and the eluted components were collected and evaporated to dryness. Water was added to prepare a solution with a specific concentration suitable for bioassay [43,44,45,46].

(c) Determination of biological activity: Similar to the method employed for the determination of toxic components of mycotoxin in Section 2.2.3, the solution obtained from the previous step was sprayed onto leaves, and the incidence of leaf lesions was observed and recorded.

#### 2.3.3. Preparation of HPLC

HPLC was performed using an LCQ Deca XP MAX instrument (Finnigan, CA, USA) equipped with an evaporative light scattering detector (ELSD). The chromatographic column employed was the Micropak NH2-10 (300 × 4 mm). Spectrum recording and data processing were carried out utilizing the chromtek chromatographic workstation. The chromatographic conditions selected for the analysis were as follows: a C18 reversed-phase column ODS-2 sized 460 mm × 250 mm with a particle size of 5 μm. The mobile phase consisted of 100% methanol, which was degassed by ultrasonic treatment for 20 min through a 0.45 μm filter membrane. The flow rate utilized was 0.4 mL/min, and the injection volume was set at 20 μL. The detection wavelength was specifically set at 280 nm, and the analysis was carried out at room temperature. For subsequent analysis, the collected mycotoxins were dissolved in methanol, while the eluent methanol was subjected to degassing through ultrasonication after vacuum filtration for further use as the mobile phase.

#### 2.3.4. UV-Visible Spectrophotometer Scanning

A small amount of the purified substance was dissolved in methanol and subjected to scanning across the entire wavelength range using a UV-2600 UV-visible spectrophotometer (Shimadzu, kyoto, Japan). Methanol was used as the blank control, and measurements were performed using quartz cuvettes.

#### 2.3.5. Purity Detection of Active Components

(a) TLC detection: The mycotoxin isolated in Section 2.3.3 was dissolved in methanol. Subsequently, TLC was conducted using the optimal developer selected in Section 2.3.2. To preliminarily confirm the presence of a single substance, the TLC plate was exposed to iodine vapor for color development. (b) HPLC detection: The isolated mycotoxin was dissolved in acetonitrile and its purity was determined using an HPLC Waters 2695 2996 detector (Waters, Milford, MA, USA). The analysis was performed at the maximum absorption ultraviolet wavelength. Parameters for the detection included an injection flow rate of 10 μL/min and a detection time of 60 min.

#### 2.3.6. Structure Identification and Biological Activity Determination of Active Components

Nuclear magnetic resonance (NMR): A suitable amount of PK mycotoxin was dissolved in deuterated DMSO and subjected to NMR analysis using a Bruker-400 MHz (Brucker, Billerica, MA, USA) superconducting nuclear magnetic resonance instrument. For both 1H-NMR and 13C-NMR analysis, tetramethylsilane tetramethylsilane (TMS) was used as the internal standard. Mass spectrometry (MS): For injection, the PK mycotoxin sample was directly injected through an injection pump into a Waters H-Class UPLC system coupled with an AB Sciex API 4000B triple quadrupole mass spectrometer (Waters, USA), was used to inject the sample through an injection pump directly. The mass spectrum of the mycotoxin, in positive ion mode, was obtained using the electrospray ion source with a spray voltage of 1000 kV. The injection pump employed an injection volume of 10 μL/min.

To evaluate biological activity, *Z. schinifolium* pots exhibiting uniform growth without any pests or diseases were selected. The pure mycotoxin was sprayed onto the selected leaves at a dosage of 1 mL per leaf. The treated leaves were then covered with absorbent cotton soaked in sterile water and wrapped with cling film. Each treatment included 10 plants of *Z. schinifolium* with 3 branches per plant and 5 leaves per branch. This experimental design was repeated 3 times to observe the incidence. Leaves treated with sterile water and a pathogen spore suspension at a concentration of 1 × 10^6^ cfu/mL were used as control samples.

### 2.4. Determination of Physiological and Biochemical Indexes of Z. schinifolium after Mycotoxin Treatment

#### 2.4.1. Sample Pretreatment

In total, 5 leaves from each *Z. schinifolium* pot plant should be disinfected with 75% alcohol, followed by rinsing with sterile water three times. Subsequently, the leaves will be sprayed with mycotoxin solutions of 10, 20, 40, and 80 μg/mL. To maintain moisture, gauze and fresh-keeping film soaked in sterile water will be used. As a control, a group of leaves treated with sterile water and a pathogen spore suspension of 1 × 10^6^ cfu/mL will be included. After specific durations of treatment (5, 8, 12, 18, 25, and 35 days), leaf tissue samples measuring 2 mm × 2 mm around the lesions will be collected for determining physiological indexes. Each index determination will be performed with 10 replicates. The recorded data will include the percentage of lesion area about the total leaf area. The same method used for Section 2.2.1 will be applied. Furthermore, correlation coefficients will be calculated to examine the relationships between physiological and biochemical indexes, time, mycotoxin concentration, and the proportion of the lesion area.

#### 2.4.2. Determination of Soluble Protein Content

To measure the soluble protein content, the Coomassie brilliant blue G-250 method was employed [47]. The solution’s optical density was recorded at 595 nm using a spectrophotometer. Using a standard curve, the protein concentration was determined; subsequently, the total protein content per gram of leaf was calculated.

#### 2.4.3. Determination of Soluble Sugar Content

The determination of soluble sugar content involved the use of the anthrone method. Absorbance was measured at 620 nm using a spectrophotometer [48]. Sugar values were obtained by referencing a standard curve.

#### 2.4.4. Determination of Chlorophyll Content

Chlorophyll content was determined through an extraction method [49]. A 10 mL solution of extractant (acetone/ethanol/distilled water = 4.5:4.5:1) was used to extract chlorophyll at a temperature of 25 °C for 24 h. Absorbance at 645 nm and 663 nm was measured using a spectrophotometer.

#### 2.4.5. Determination of MDA Content

The measurement of MDA content was measured conducted using the thiobarbituric acid method [49].

#### 2.4.6. Effect of Mycotoxin on Plant Enzyme Activities

(a) Determination of SOD activity: SOD activity was determined using the NBT photoreduction method [50]. Absorbance at 560 nm was measured using a spectrophotometer, and the specific activity was calculated based on the 50% inhibition of NBT photoreduction. (b) Determination of CAT activity: CAT activity was measured by monitoring the decrease in absorbance at a wavelength of 240 nm. CAT was used to decompose hydrogen peroxide, and the rate of absorbance change was measured according to the reaction time [51].

### 2.5. Statistical Analyses

The collected data were analyzed using one-way ANOVA in the SPSS software (version 27.0 for Windows, SPSS Inc., Chicago, IL, USA). Significance was determined at *p* < 0.05. To assess the correlation between physiological and biochemical indexes, time, mycotoxin concentration, and lesion area ratio, the Tutools platform (https://www.cloudtutu.com) was utilized, which was accessed on 10 August 2023. Pearson’s correlation coefficients and significance levels were calculated for this analysis. * 0.01 < *p* < 0.05; ** 0.001 < *p* ≤ 0.01; *** *p* ≤ 0.001.

## 3. Results

### 3.1. Screening of the Best Medium for P. kenyana to Produce Mycotoxin and Determination of the Toxin Components of Fungal Secondary Metabolites

The toxicity test revealed that the metabolites of *P. kenyana*, when cultured in four different media, induced yellow/brown lesions on the leaves of *Z. schinifolium*. However, their toxicity varied (Figure 1). Among the four media, the metabolites of the fungus cultures in PDB resulted in the highest pathogenicity, resulting in a lesion area that accounted for 57.17% of the total leaf area. The metabolites of the fungus cultures in modified martin broth, potato sucrose broth, and czapek dox broth exhibited lower toxicity, with lesion areas accounting for 23.10%, 20.68%, and 19.38% of the total leaf area, respectively. Therefore, PDB was identified as the most suitable medium for mycotoxin production by *P. kenyana*. Consequently, *P. kenyana* was cultured in PDB at 28 °C and 150 r/min shaking for 18 days in subsequent experiments.

Ammonium sulfate fractional precipitation of the incubation broth yielded non-protein (supernatant) and protein (precipitate) substances at various saturation levels, which were tested for toxicity (Appendix A and Figure 2). The results showed that the non-protein fraction in the incubation broth was active, whereas the protein fraction was inactive. Therefore, it can be concluded that the active substances responsible for toxicity are non-proteins.

### 3.2. Isolation and Purification of Mycotoxin

#### 3.2.1. The Results of Thin Layer Chromatography, Silica Gel Column Chromatography, and Preparative High-Performance Liquid Chromatography

The substances were subjected to analysis using TLC, silica gel column chromatography, and preparative HPLC. During TLC analysis, it was observed that the petroleum ether/ethyl acetate (*v*/*v*) system provided better separation. Figure 3 illustrates that the ratios of 2:1 and 3:1 for petroleum ether/ethyl acetate could separate up to six substances, with the 3:1 ratio resulting in the best separation and expansion (Figure 3A). Therefore, the 3:1 ratio was selected for elution in the silica gel column chromatography analysis. The eluted fractions were collected, and similar components with the same Rf values were combined, separated, and purified using preparative HPLC (Figure 3B).

The mycotoxin isolated through column chromatography was designated as PK mycotoxin. During the initial purification process via column chromatography, four separate components were obtained, as follows: component I (8.32–8.61 min), component II (17.48–17.78 min), component III (20.22–20.55 min), and component IV (27.12–27.49 min). Component III had a significantly lower content, and TLC analysis did not reveal a clear main point. Components I, III, and IV, which met the requirements for further purification, were named PK-1, PK-2, and PK-3, respectively. These components were dissolved in acetonitrile and subjected to purification using the UV-3000 preparative HPLC system (Beijing Tong Heng Innovation Technology Co., Ltd., Beijing, China). The chromatographic conditions consisted of a C18 preparative column (20 × 250 mm) with static axial compression, 72% acetonitrile as the mobile phase, a detection wavelength of 275 nm, and a flow rate of 1 mL/min. Finally, pure PK mycotoxin (PK-1, PK-2, and PK-3) was obtained.

#### 3.2.2. UV-Visible Spectrophotometer Scanning

PK-1, PK-2, and PK-3 were examined using a UV-visible spectrophotometer, with methanol serving as the control. The results depicted in Figure 4 demonstrated that PK-1 exhibited peaks at 202.7 nm, 247.3 nm, 268.6 nm, and 330.4 nm, with the highest absorption occurring at 202.7 nm. PK-2 exhibited peaks at 221.4 nm and 308.9 nm, with the highest absorption occurring at 221.4 nm. PK-3 exhibited peaks at 221.4 nm and 275.7 nm, with the highest absorption also occurring at 221.4 nm. Based on these findings, the detection wavelengths of 202.7 nm, 221.4 nm, and 221.4 nm were selected for PK-1, PK-2, and PK-3, respectively.

#### 3.2.3. Purity Detection of Active Components

The purity of the isolated compounds PK-1, PK-2, and PK-3 was evaluated using TLC and HPLC. TLC analysis, combined with UV irradiation, confirmed that all three compounds exhibited a single point, indicating their purity as shown in Appendix A. To further assess their purity, HPLC was employed, with detection wavelengths of 202.7 nm, 221.4 nm, and 221.4 nm for PK-1, PK-2, and PK-3, respectively. The HPLC analysis revealed that the mycotoxins had high purity levels. Specifically, the main peak of PK-1 exhibited a purity of 97.79%, the main peak of PK-2 had a purity of 95.35%, and the main peak of PK-3 had a purity of 99.49%, as depicted in Figure 5. All three mycotoxins exhibited purities exceeding 95%, rendering them suitable for further structural analysis.

#### 3.2.4. Structure Identification and Biological Activity Determination

Based on a comprehensive analysis of the results presented in Figure 6A–F, the following determinations can be determined: PK-1 has a molecular mass of 402.39 and a molecular formula of C_21_H_22_O_8_, and its chemical name is Nobiletin. PK-2 has a molecular mass of 138.07 and a molecular formula of C_8_H_10_O_2_, and its chemical name is p-Hydroxyphenethylalcohol. PK-3 has a molecular mass of 180.20 and a molecular formula of C_10_H_12_O_3_, and its chemical name is pestalopyrone. The structural formulas of PK-1, PK-2, and PK-3 are displayed in Figure 6G–I.

After conducting separate applications of the isolated PK-1, PK-2, and PK-3 compounds on *Z. schinifolium* pots. It was observed that PK-1 and PK-2 did not have any toxic effects on *Z. schinifolium*. However, the application of PK-3 resulted in noticeable symptoms on *Z. schinifolium*. These symptoms manifested as yellow/brown lesions that closely resembled the symptoms of the natural disease (Figure 6J–L). These findings revealed the significant toxicity of PK-3 on the leaves of *Z. schinifolium*, suggesting that it is the primary mycotoxin.

### 3.3. Determination of Physiological and Biochemical Indexes of Z. schinifolium after Mycotoxin Application

The effects of the PK-3 mycotoxin on the physiological and biochemical indexes of *Z. schinifolium* plants were investigated. The results revealed that the protein and MDA content in the leaves increased progressively within 35 days after the application of the mycotoxin. Conversely, the soluble sugar and chlorophyll content showed a gradual decrease over time. The activities of SOD and CAT initially exhibited an increase but subsequently declined (Figure 7).

Appendix A illustrates the incidence of *Z. schinifolium* after treatment with PK-3 mycotoxin. By calculating the percentage of lesion area relative to the total leaf area of *Z. schinifolium* (Figure 8A), it was observed that PK-3 toxicity and *P. kenyana* pathogenicity had similar effects on the plant. Within 35 days of treatment with mycotoxin, the proportion of lesion area differed significantly from the control treated with sterile water, and this proportion increased with higher concentrations of mycotoxin.

Figure 8B presents the correlation between physiological and biochemical indexes, time, mycotoxin concentration, and the proportion of lesion area. These results further highlight the relationship between the physiological and biochemical indexes of *Z. schinifolium* and factors such as time and mycotoxin concentration. Most of the physiological and biochemical parameters exhibited varying degrees of correlation with time, mycotoxin concentration, and the proportion of lesion area. Notably, time displayed significant positive correlations with MDA content (*p* < 0.01), soluble protein concentration (*p* < 0.001), and the proportion of lesion area (*p* < 0.001). Conversely, time exhibited a significant negative correlation with CAT activity (*p* < 0.01). The mycotoxin concentration displayed significant positive correlations with the soluble protein concentration (*p* < 0.001), MDA content (*p* < 0.001), and SOD activity (*p* < 0.01). Conversely, it showed significant negative correlations with soluble sugar content (*p* < 0.001) and chlorophyll content (*p* < 0.001).

## 4. Discussion

The potato glucose medium is a commonly used fungal medium for the cultivation and separation of fungal metabolites. It was observed that *P. microspora* exhibited the fastest growth on the potato dextrose agar (PDA) and potato sucrose agar (PSA) [52]. *Pestalotiopsis* sp. BC55 produced the highest amount of exopolysaccharide when grown in PDB supplemented with glucose [53], after a mixed culture of *Nigrospora oryzae* TPY 10-1 and *Irpex lacteus* ZPY 45-2 in PDB. From the culture medium and mycelium, a total of 14 compounds were isolated and identified. According to the results of the toxicity test in this study, the metabolites of *P. kenyana* cultured in PDB exhibited the largest lesion area on *Z. schinifolium* leaves, indicating the highest toxicity. Therefore, among the four tested media, PDB was selected as the ideal medium for subsequent culture due to its ability to produce metabolites with the strongest toxicity.

Mycotoxins can be classified into various chemical categories, such as proteins, esters, ketones, sugars, organic acids, and alkaloids. Before separating and purifying mycotoxins, it is necessary to determine whether they are proteins or not. Different separation methods should be adopted accordingly. In this experiment, it was found that the mycotoxins from *P. kenyana* are non-protein substances. Organic solvents are often used to extract and enrich the fermentation broth, which is then detected and separated through chromatographic techniques. For instance, the ethyl acetate extraction method was employed to isolate ten compounds from the solid culture of *P. sydowiana*. Reversed-phase high-performance liquid chromatography was utilized in this process [54]. The fermentation broth of *P. karstenii* was subjected to silica gel CC using petroleum ether and ethyl acetate, leading to the isolation of four compounds [55]. In this experiment, multiple extractions and enrichments were conducted using ethyl acetate to obtain crude mycotoxin. The crude toxin from *P*. *kenyana* was then purified through silica gel CC and HPLC, utilizing a petroleum ether and ethyl acetate mixture in a 3: 1 ratio. As a result, three compounds were obtained. However, the limited sample size, possible failure to separate low-content components, or improper operation during the crude toxin preparation and chromatography process may have accounted for the low number of isolated compounds. In addition, the complexity of the mycotoxin components sometimes requires the use of multiple developing agent systems. The structure of the separated mycotoxins needs to be further analyzed using techniques like NMR and MS. For example, compounds isolated from the culture of *P. theae* were identified as Pestalazines A and B, and Pestalamides A–C using NMR spectroscopy [56]. The structures of these three compounds were examined through NMR hydrogen and carbon spectra in our experiment.

One of the compounds that was isolated in this experiment, PK-3 (pestalopyrone), has been previously reported as a fungal metabolite and mycotoxin. It was identified from the extracts of the mangrove endophytic fungus *Nigrospora oryzae*, along with Sterigmatocystin [57]. Pestalopyrone, which is a mycotoxin produced by *P. oenotherae*, is 6-(1′-methylprop-1′-enyl)-4-methoxy-2-pyrone. Additionally, it was also isolated from the culture filtrate of *P. guepinii* [22]. Compared with the previous studies on the pathogenic fungi of *Pestalotiopsis* sp., the same metabolites were isolated from *P*. *kenyana* in this experiment. These findings suggested that pestalopyrone may serve as a virulence factor produced by the pathogenic fungi of *Pestalotiopsis* sp., which induces pathogenic fungi to cause pathogenic effects on host plants. Therefore, it can be speculated that pestalopyrone is a toxic substance produced by *P*. *kenyana*.

The evaluation of soluble sugar and soluble protein content is crucial in assessing the plant defense system. Recent studies have suggested that the soluble sugar content might be associated with host disease resistance [58]. For example, cold stress has been shown to cause a significant reduction in the soluble sugar content of *Eupatorium adenophorum* [59]. Similarly, when soybean roots were treated with the *F. oxysporum* mycotoxin, when or pecans were infected with *Phomopsis* spp., there was a decrease in soluble sugar content [60]. In our experiment, we observed a gradual decrease in the soluble sugar content of *Z. schinifolium* leaves over time following treatment. The application of a mycotoxin led to an increase in *Z. schinifolium* respiration and the consumption of soluble sugar.

The soluble protein content tends to increase under stressful conditions, as it aids the host plant in resisting further pathogenic infections by promoting protein synthesis and accumulation. For instance, spring wheat has been found to exhibit an increase in total soluble protein content under salt stress [61]. In our experiment, we observed that the protein content in *Z. schinifolium* leaves increased over time following treatment, suggesting that the mycotoxin induced protein accumulation.

The measurement of MDA content in affected plant tissues is commonly used as an indicator of cell membrane damage under stress conditions. Numerous studies have demonstrated a negative correlation between MDA content and plant disease resistance, where higher content indicates more severe cell damage [62]. In our experiment, we observed an increase in the MDA content of *Z. schinifolium* leaves over time following the treatment, indicating membrane lipid peroxidation and an escalation in cell damage.

Chlorophyll is an essential component for plant photosynthesis, and its content plays a vital role in determining plant disease resistance. Pathogen infection can lead to the destruction of chloroplasts or the inhibition of chlorophyll synthesis. Generally, higher chlorophyll content is associated with greater disease resistance [63]. When plants are infected by pathogenic fungi, there is often a significant decrease in their chlorophyll content. For instance, the total chlorophyll and carotenoid contents of *Brassica juncea* L. var. Pusa bold seeds decreased by 67.61% and 82.70%, respectively, when treated with 2000 g/L aflatoxin B1 [64]. Similarly, in corn, the total chlorophyll, chlorophyll a, and chlorophyll b content decreased by 49.4%, 38.6%, and 64.1%, respectively, after treatment with toxic compounds from *Exserohilum turcicum* [65]. In our experiment, we observed a significant decrease in the chlorophyll content of *Z. schinifolium* leaves over time following treatment. This suggests a destructive effect on chloroplasts or inhibitory effects on chlorophyll synthesis induced by the mycotoxin.

When plants are infected by pathogens, they undergo various physiological responses to induce systemic resistance. In particular, key enzymes such as SOD, CAT, POD, and PAL are of interest as indicators of plant-induced resistance [66,67]. In our study, we investigated the activity of SOD and CAT in *Z. schinifolium* leaves after the application of mycotoxins. We observed that these activities initially increased and then decreased over time following mycotoxin application. This indicated the activation of *Z. schinifolium*’s antioxidant mechanism and the induction of defense enzyme activity cause by the pathogenic infection. Additionally, the protein content and MDA content in the leaves increased with treatment time for up to 35 days after mycotoxin application, as indicated by the physiological and biochemical indexes. Conversely, the soluble sugar content and chlorophyll content decreased over time, with the chlorophyll content reaching near-zero levels. The activities of SOD and CAT initially increased and then declined over time. Overall, our results demonstrated that *Z. schinifolium* exhibited a series of physiological responses in its defense system after pestalopyrone application, supporting the notion that *P. kenyana* caused *Z. schinifolium* leaf spot through the production of pestalopyrone.

## 5. Conclusions

In this study, it was determined that PDB was the most favorable medium for *P. kenyana* for the production of mycotoxins. Additionally, the non-protein mycotoxin PK-3 exhibited significant toxicity in response to *Z. schinifolium*. This finding suggests that pestalopyrone, which is produced by *P. kenyana*, plays a pivotal role in inducing the pathogenicity of *P. kenyana* to *Z. schinifolium*. The insights gained from this study extend our understanding of pestalopyrone and pave the way for further research on the pathogenic mechanism of *Z. schinifolium* leaf spot, which is caused by *P. kenyana*.

## Figures and Tables

**Figure 1 jof-09-01194-f001:**
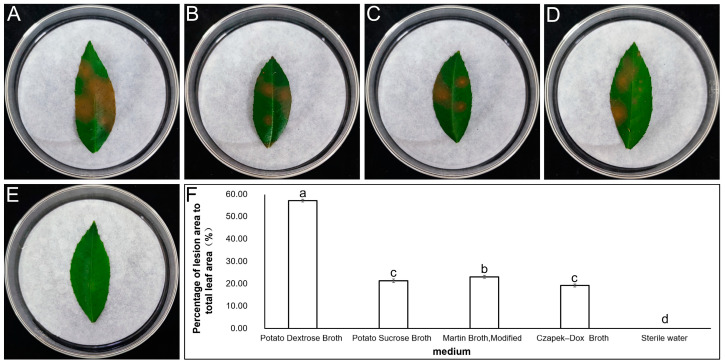
The effect of metabolites of *P. kenyana* cultivated in different media on the lesion area of *Z. schinifolium*. Note: (**A**) potato dextrose broth; (**B**) potato sucrose broth; (**C**) martin broth, modified; (**D**) czapek dox broth; (**E**) sterile water; (**F**) *Z. schinifolium* lesion area and significance of differences. (**A**–**E**) show the toxicity of metabolites of *P. kenyana* cultured in different media to *Z. schinifolium*. The yellow/brown spots in the images represent the lesions caused by the fungal metabolites of *P. kenyana*. The data presented in (**F**) show the percentage of lesion area to total leaf area (%) and are the average of 10 replicates. Different lowercase letters indicate significant differences in the lesion area on *Z. schinifolium* leaves caused by the metabolites of *P. kenyana* cultivated in different media at a significance level of *p* < 0.05 (LSD method).

**Figure 2 jof-09-01194-f002:**
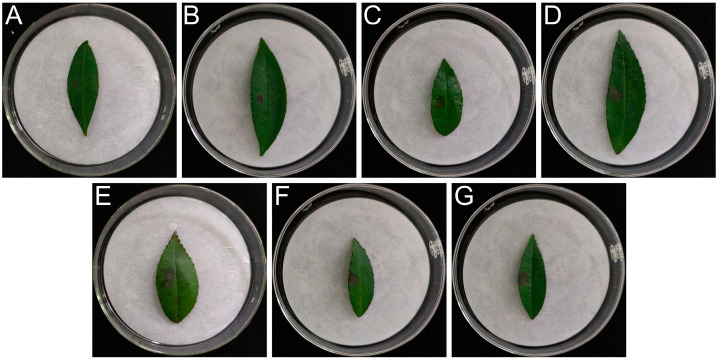
Toxicity test results of ammonium sulfate precipitation supernatant. Note: (**A**) 20% supernatant; (**B**) 30% supernatant; (**C**) 40% supernatant; (**D**) 50% supernatant; (**E**) 60% supernatant; (**F**) 70% supernatant; (**G**) 80% supernatant.

**Figure 3 jof-09-01194-f003:**
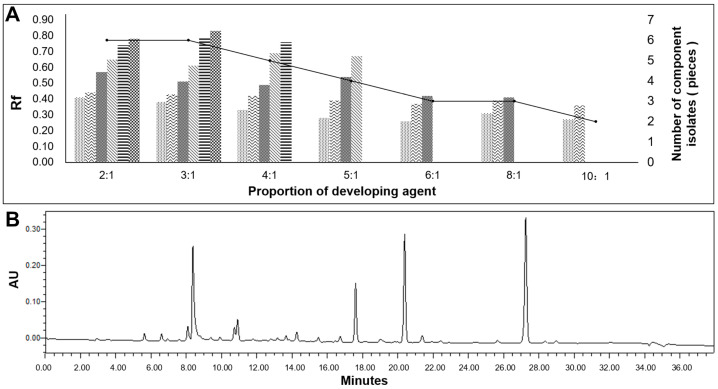
Separation and purification. Note: (**A**) TLC results; (**B**) high-performance liquid chromatograph test results. Different patterns in (**A**) represent different substances separated.

**Figure 4 jof-09-01194-f004:**
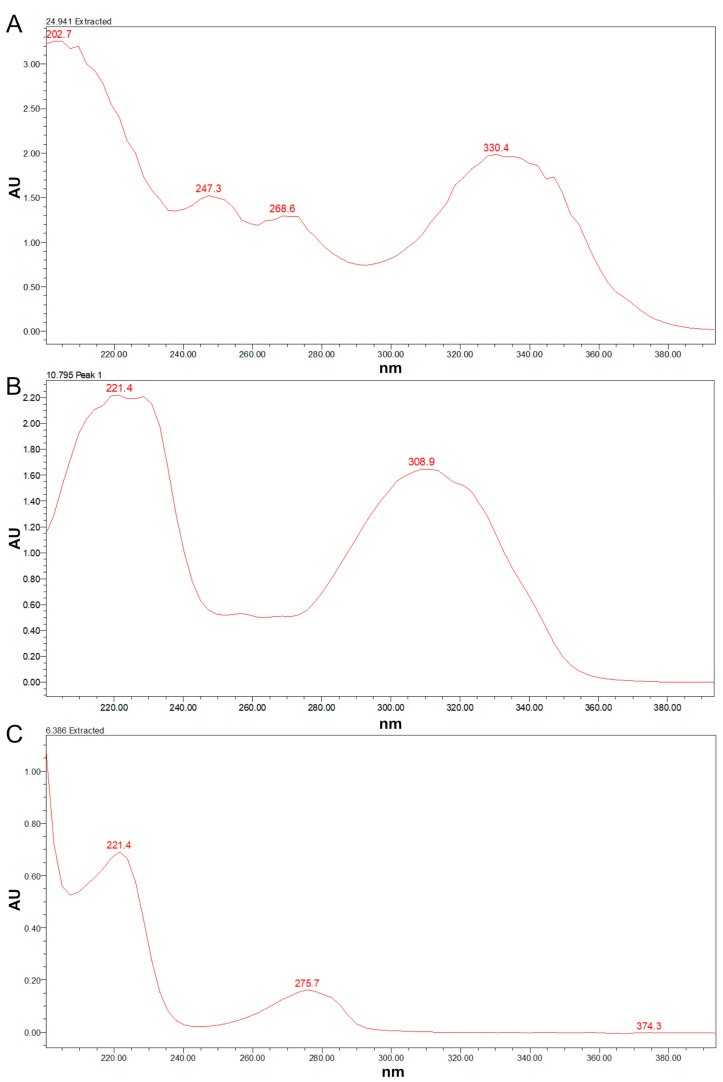
UV-visible spectrophotometer scanning diagram. Note: (**A**) PK-1; (**B**) PK-2; (**C**) PK-3.

**Figure 5 jof-09-01194-f005:**
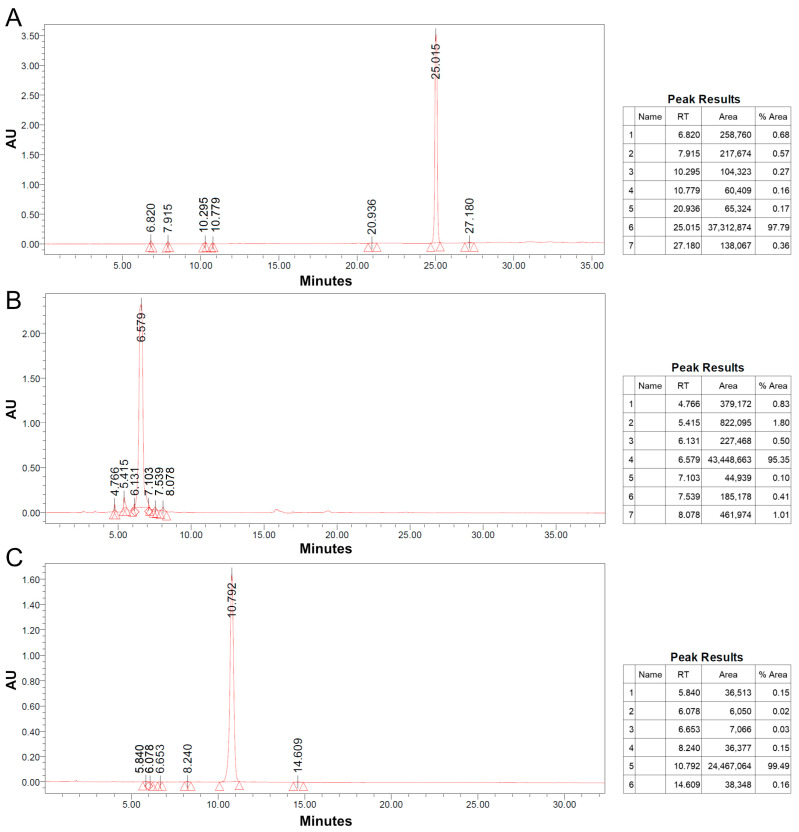
HPLC detection diagram. Note: (**A**) PK-1; (**B**) PK-2; (**C**) PK-3.

**Figure 6 jof-09-01194-f006:**
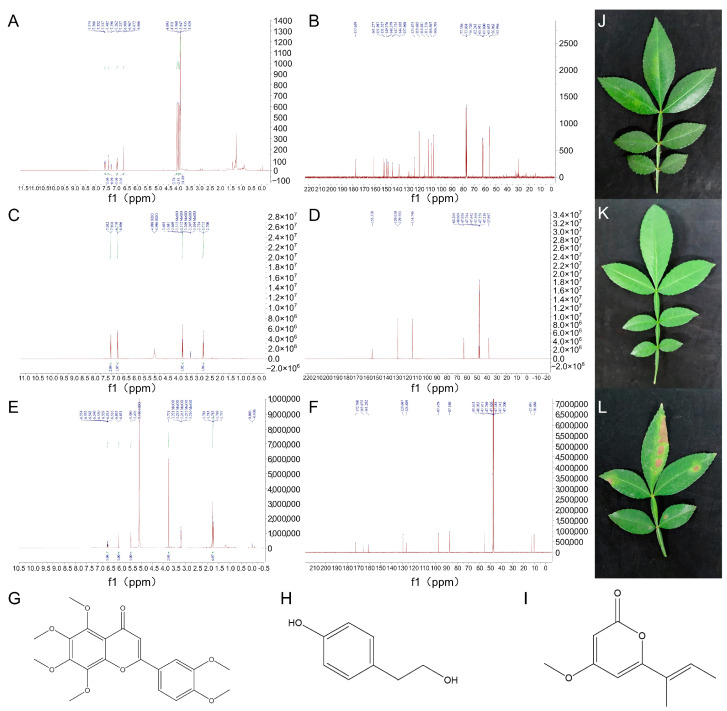
Nuclear magnetic resonance carbon spectrum and hydrogen spectrum; determination of mycotoxin structure and biological activity. Note: (**A**) PK-1 carbon spectrum; (**B**) PK-1 hydrogen spectrum; (**C**) PK-2 carbon spectrum; (**D**) PK-2 hydrogen spectrum; (**E**) PK-3 carbon spectrum; (**F**) PK-3 hydrogen spectrum; (**G**–**I**) structural formula. (**G**) PK-1; (**H**) PK-2; (**I**) PK-3; and parts (**G**,**H**) are the results of the bioactivity determination of treated leaves on *Z. schinifolium* pot. (**J**) PK-1; (**K**) PK-2; (**L**) PK-3.

**Figure 7 jof-09-01194-f007:**
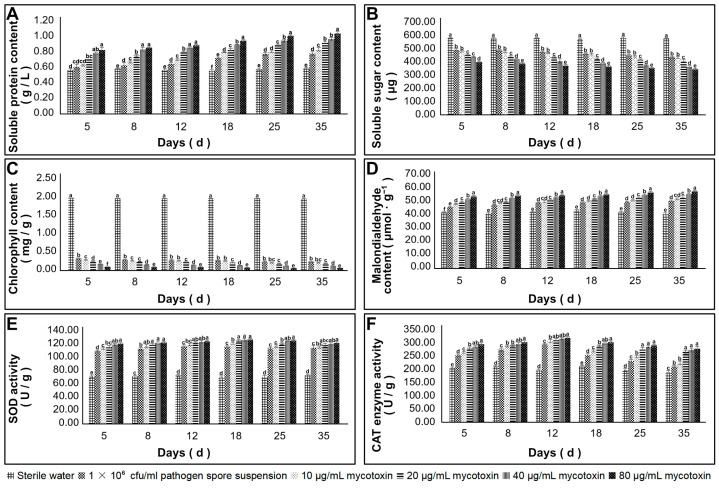
Effects of mycotoxin application on physiological and biochemical indexes of *Z. schinifolium.* Note: (**A**) soluble protein concentration; (**B**) soluble sugar content; (**C**) chlorophyll content; (**D**) MDA content; (**E**) SOD enzyme activity; (**F**) CAT enzyme activity. Different lowercase letters are used to denote significant differences in the effects of various treatments on the physiological and biochemical indexes of *Z. schinifolium*, with a significance level set at *p* < 0.05 (LSD method).

**Figure 8 jof-09-01194-f008:**
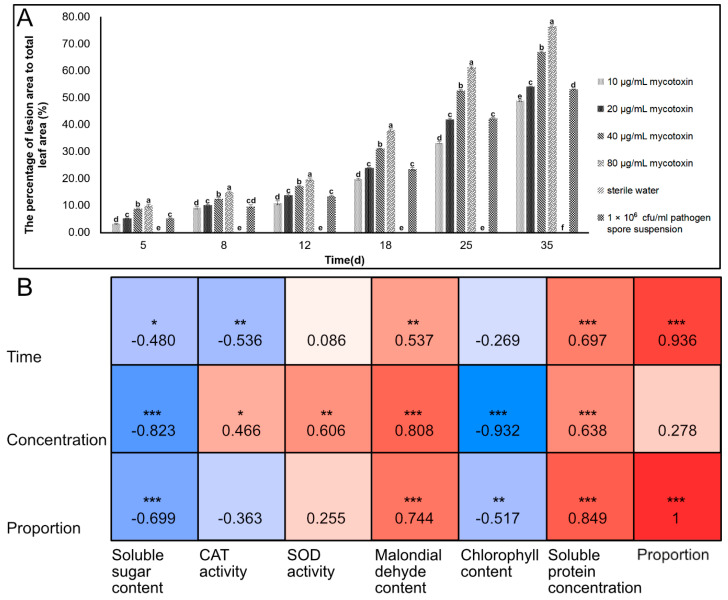
The percentage of lesion area to total leaf area and correlation analysis of *Z. schinifolium* treated with PK-3 mycotoxin. Note: (**A**) the percentage of lesion area to total leaf area of *Z. schinifolium* leaves treated with PK-3 mycotoxins; (**B**) the correlation between the proportion of lesions and time, mycotoxin concentration, and physiological and biochemical indexes of *Z. schinifolium* after PK-3 mycotoxin treatment. Different lowercase letters in (**A**) were used to indicate the significant difference in the percentage of lesion area to total leaf area of *Z. schinifolium* caused by different concentrations of toxins, and the significant level was set to *p* < 0.05 (LSD method). Time: the time after *Z. schinifolium* inoculation of PK-3 mycotoxin. Concentration: the concentration of PK-3 mycotoxin. Proportion: the percentage of lesion area to the total leaf area of *Z. schinifolium*. * 0.01 < *p* < 0.05; ** 0.001 < *p* ≤ 0.01; *** *p* ≤ 0.001. In figure (**B**), red represents positive correlation and blue represents negative correlation.

## Data Availability

Data are contained within the article and Appendix A.

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
