# Peer review of "The Identification and Role of the Key Mycotoxin of Pestalotiopsis kenyana Causing Leaf Spot Disease of Zanthoxylum schinifolium"

_jof, 2023, doi:10.3390/jof9121194_

Round 1

Reviewer 1 Report

Comments and Suggestions for Authors

Dear Authors,

Please find the following comments that have to be taken into account in the entire manuscript:

The introduction has a lot of theoretical information that are not relevant to the topic of the work. It is very exhaustive and has to be shortened.

The term “Toxin” was used to indicate “Mycotoxin”. Please replace it.

The term “pathogenicity” was used to refer to the mycotoxin, and this is wrong. This term is used to refer to living organisms like fungi. It must be replaced with “toxicity”.

The term “fermentation” is mistakenly used. The fermentation occurs in the absence of oxygen while in the case of this fungus, the oxygen exists. Please replace it with “incubation”.

The term “toxin inoculation” must be replaced with “mycotoxin application”.

The term “toxin infected” must be replaced with “mycotoxin treated”.

I underlined all these terms in the manuscript. Please take care of them.

In addition, Please take care of all these remarks in the supplementary materials as well.

The conclusion is Too short

In addition, please find my detailed comments in the attached pdf file.

Good luck

Comments on the Quality of English Language

The usage of pathological terms.

Author Response

Modification instructions

Dear Reviewer,

Thank you for your comments concerning our manuscript entitled "Identification and role of the key mycotoxin of Pestalotiopsis kenyana, causing leaf spot disease of Zanthoxylum schinifolium " (Manuscript ID: jof-2717955).

Those comments are all valuable and very helpful for revising and improving our paper, as well as the important guiding significance to our research. Revised portions are marked in the paper. The main corrections in the paper are as follows:

To question 1: P1L2 The term pathogenic is used for the fungus not for the toxin reformulate and please use mycotoxin instead. Please take care of this comment in the entire manuscript.

Answer: We appreciate the reviewer’s important comment. We have changed the title to “Identification and role of the key mycotoxin of Pestalotiopsis kenyana, causing leaf spot disease of Zanthoxylum schinifolium.”. In addition, we have removed pathogenicity and changed toxins into mycotoxins in the full text.

To question 2: P1L20 how did you do this test?

Answer: We appreciate the reviewer’s important comment. We have changed the sentence to “The biological activity test by spray method showed that PK-3 had strong toxicity and was presumed to be the main mycotoxin.”.

To question 3: P1L22-24 Please clarify this sentence.

Answer: We appreciate the reviewer’s important comment. We have changed the sentence to “The results showed that the protein content and malondialdehyde content of leaves increased with the increase of time within 35 days after mycotoxin treatment.”.

To question 4: P1L25 what do you mean with this?

Answer: We appreciate the reviewer’s important comment. Treatment time refers to the time after application of PK-3 mycotoxins to Z.schinifolium.We have deleted the treatment.

To question 5: P1L27-28 The toxin could be virulence factor that help the pathogen to induce the infection. Please reformulate this sentence.

Answer: We appreciate the reviewer’s important comment. We have changed the sentence to “Therefore, it is believed that the mycotoxin Pestalopyrone could be a virulence factor that helps P. kenyana to induce the infection of Z. schinifolium.”.

To question 6: P1L29-30 This is a general sentence. Please be more specific and try to reformulate this sentence in scientific way.

Answer: We appreciate the reviewer’s important comment. We have changed the sentence to “This study provides a theoretical basis for exploring the pathogenic mechanism of Z. schinifolium leaf spot, lays a foundation for better prevention and control, and enriches the research of mycotoxins in China”.

To question 7: P1L31-32 these are already mentioned in the title. please find other keywords.

Answer: We appreciate the reviewer’s important comment. We have deleted Zanthoxylum schinifolium and Pestalotiopsis kenyana, and changed keywords to “Prickly ash, Mycotoxins, Pestalopyrone, Toxicity”.

To question 8: P1L43-P2L45 These are very selective examples. please remove them.

Answer: We appreciate the reviewer’s important comment. We have deleted the sentence.

To question 9: P2L56-80 These are theoretical information that are not relevant to your study.Are these metabolites contribute to the pathogenicity of these fungi?

You are mentioning secondary metabolites that are produced by different fungi of Pestalotiopsis sp. as introduction to your work, however, these metabolites do not contribute in the pathogenicity of these fungi. Please try to concentrate on the metabolites that are fundamental in fungus pathogenicity and show this in your literature review.

Answer: We appreciate the reviewer’s important comment. We have deleted the contents of secondary metabolites produced by different fungi of Pestalotiopsis and added pathogenicity of these fungi “The pathogenic fungus of tea gray blight fungus, P. longiseta, can produce two mycotoxins, oxysporone and ( + ) -epiepoxydon. In addition, P. microspora and P. funereacan can also produce mycotoxins and infect plants.”.

To question 10: P2L83 this means absence of oxygen. Is this right in this case?Is it fermentation or incubation?

Answer: We appreciate the reviewer’s important comment. We have changed it to incubation and modified the fermentation in the full text.

To question 11: P2L84 reformulate

Answer: We appreciate the reviewer’s important comment. We have changed the sentence to “their mycotoxins are often secreted into the culture medium.”.

To question 12: P2L89-90 improve the language

Answer: We appreciate the reviewer’s important comment. We have changed the sentence to “among which the type of medium has the greatest impact on the number of mycotoxins”.

To question 13: P3L105-110 Theoretical information

Answer: We appreciate the reviewer’s important comment. We have deleted this content.

To question 14: P3L120-141 this paragraph is not relevant to your study.

Answer: We appreciate the reviewer’s important comment. We have deleted the irrelevant content.

To question 15: P4L154 the mycotoxin does not cause any disease, the disease is caused by the fungus itself.

Answer: We appreciate the reviewer’s important comment. We have changed the sentence to “Finally, through the determination of toxicity and physiological and biochemical indicators, it was determined whether mycotoxin caused the lesions of Z. schinifolium, and then whether mycotoxins are virulence factors that help P. kenyana to induce infection, to provide a theoretical basis for the prevention and control of the leaf spot disease of Z. schinifolium.”.

To question 15: P4L169-174 I think that you filled with water to reach final volume of 1000 mL?If I am right please correct the description and take care of it.

Answer: We appreciate the reviewer’s important comment. We have changed the sentence to “and filled with distilled water to reach a final volume of 1000 mL.”, and we modified the rest of the full text.

To question 16: P4 L180is it 6mm2?

Answer: We appreciate the reviewer’s important comment. It is not 6mm2. We have changed the sentence to “a 6mm diameter fungus pieces”.

To question 17: P7L299This wavelength is in the visible range not in the UV range.

Answer: We appreciate the reviewer’s important comment. Here the writing is wrong, we have deleted “UV”, which is the spectrophotometer. And we modified the other parts of the full text.

To question 18: P7L332-334 the fungus cultured in PDB......

Answer: We appreciate the reviewer’s important comment. We have changed the sentence to “ the fungi cultured in PDB resulted in the highest lesion area, accounting for 57.17% of the total leaf area, signifying the strongest pathogenicity. The fungi cultured in modified martin broth, potato sucrose broth, and czapek dox broth exhibited lesser toxicity,”.

To question 18: P8L341 Please clarify

Answer: We appreciate the reviewer’s important comment. We have changed the sentence to “The effect of P. kenyana cultivated in different media on the lesion area of Z. schinifolium. Note: A, Potato dextrose broth; B, Potato sucrose broth; C, Martin broth, modified; D, Czapek dox broth; E, Sterile water; F, Z. schinifolium lesion area and significance of differences.”.

To question 19: P15L457-460 reformulate

Answer: We appreciate the reviewer’s important comment. We have changed the sentence to “Figure 8. The incidence and correlation analysis of Z. schinifolium treated with PK-3 mycotoxin. Note: A, The proportion of Z. schinifolium lesion area to total leaf area after PK-3 mycotoxin treatment; B, Correlation analysis of the proportion of lesions of Z. schinifolium treated with PK-3 mycotoxin with time, mycotoxin concentration and physiological and biochemical indexes.”.

To question 20: P15L464 Please separate them. Start with discussion and then conclusions.

Answer: We appreciate the reviewer’s important comment. We have separated the discussion and conclusions 

To question 21: P16L488 the medium does not produce mycotoxin, it is produced by the fungus itself

Answer: We appreciate the reviewer’s important comment. We have changed the sentence to “Therefore, among the four tested media, P. kenyana cultured in PDB had the strongest toxicity and was used for subsequent culture.”

To question 22: P17L559 Too short conclusion

Answer: We appreciate the reviewer’s important comment. We have added the conclusion “The physiological and biochemical indexes results showed that the protein content and MDA content of leaves increased with treatment time up to 35 days after mycotoxin application. Conversely, the soluble sugar content and chlorophyll content decreased over time. The chlorophyll content decreased significantly, approaching 0. The activities of SOD and CAT showed an initial increase followed by a decline. ”

We tried our best to improve the manuscript and made every change in the manuscript according to the reviewers’ comments. In addition, we have looked for native language experts to polish the article. These changes will not influence the content and framework of the paper.

We appreciate for editor and reviewers’ warm work earnestly, and hope that the correction will meet with approval.

Once again, thank you very much for your comments and suggestions.

Yours sincerely,

Chang Liu, Yiling Li

Reviewer 2 Report

Comments and Suggestions for Authors

Dear colleagues!

The article “Identification and role of the key pathogenic toxin of Pestalotiopsis kenyana, causing leaf spot disease of Zanthoxylum schinifolium” is provided a theoretical basis for the prevention and control of Z. schinifolium leaf spot. The mycotoxin, Pestalopyrone (PK-3), was detected. PK-3 had significant pathogenicity in Z. schinifolium leaves and was the primary pathogenic toxin of Pestalotiopsis kenyana.

The list of comments:

1. The section “Introduction”. Lines 46-100: Too wide a range of toxins for a wide range of plant species is presented. 3 toxins are investigated only in this research. Of which only one toxin significantly affected the plant. Therefore, it is advisable to narrow down the spectrum of analyzed toxins in the section “Introduction". Lines 124-141: It would be possible to shorten the overview of information about the role of malondialdehyde.

2. The section “Introduction”. Line 53: Delete repeat, “Leaf Spot of Ixora chinensis, Leaf Spot of Ixora chinensis".

3. The section “Introduction”. Line 76: Delete repeat, “by producing toxins, and the toxins".

4. The section “Introduction”. Line 84: extra dot, “medium. [35,36].”.

5. The section “Introduction". Line 100: “pathogenic bacteria”. There may be a typo, because the article is devoted to fungal pathogens.

6. The section “Introduction”. Lines 142-156: Too blurry a aim with a lot of subtasks. It would be good to state the aim more clearly.

7. The section “Materials and Methods”. Line 158: typo, “Meterials".

8. The section “Materials and Methods”. Line 165: extra dot, “University. [57]”.

9. The section “Materials and Methods”. Line 167: put a dot.

10. The section “Materials and Methods”. Line 189: “the leaves were placed in Petri dishes”. It is unclear whether the effect of processing in Petri dishes can be extrapolated to live plants.

11. The section “Materials and Methods”. Line 207: It is unclear for what reason it is “140 leaves” of Z. schinifolium were taken.

12. The section “Materials and Methods". Lines 276-277: English translation error, the phrase “Each plant selected 3 branches, each branch selected 5 leaves.”

13. The section “Materials and Methods". Line 279: possible typo, the phrase “Each treatment included 10 strains of Z. schinifolium".

14. The section “Materials and Methods”. Lines 208-209 and 278: there may be a typo in concentration, the phrase “10 µL“ and the phrase "a dose of 1mL per leaf".

15. The section “Materials and Methods”. Lines 159-296: a complex scheme of experiments. Perhaps a figure with a diagram of experiments will facilitate the understanding of the study.

16. The section “Materials and Methods”. Lines 326: you need to specify which significance levels were calculated.

17. The section “Results”. Line 345: “the average of 10 replicates”. Coordinate with the information in the section “Materials and Methods".

18. The section “Results”. Lines 356-358: The same effect at all levels of saturation of ammonium sulfate (Table 1). An uninformative table should be moved to the section “Supplementary Materials”.

19. The section “Results”. Lines 376-377: probably an error in the number of components (there are not five, but four), the phrase “Component III had very low content, and no clear main point could be determined through TLC. Components I, III, and V,”.

20. The section “Results”. Line 385-386: extra dot, “Figure. 4 showed”.

21. The section “Results”. Line 451 and 457-463 (Figure 8): It is necessary to present the values of the correlation coefficients. It is unclear whether the correlations are positive or negative.

22. The section “Results”. Line 449: slang, “time shown”.

23. The section “Conclusions and discussions”. Line 432: it is necessary to explain the reason for the very low chlorophyll content in the variants compared to the “CK1” variant, the differences are 8-16 times. Were there similar intensity effects on other plants?

24. The section “Conclusions and discussions”. Lines 490-491: incorrect grouping by category, “including peptides, proteins, sugars, lipids, aromatic rings, heterocycles, and organic acids.”

25. The section “References". Check the references. Lines 716-718: for example, the bibliographic reference [58] is incomplete.

Reviewer's conclusion: reconsider after minor revision.

Comments on the Quality of English Language

The list of comments:

1. The section “Materials and Methods”. Line 158: typo, “Meterials".

2. The section “Materials and Methods". Lines 276-277: English translation error, the phrase “Each plant selected 3 branches, each branch selected 5 leaves.”

3. The section “Results”. Line 449: slang, “time shown”.

Author Response

Modification instructions

Dear Reviewer,

Thank you for your comments concerning our manuscript entitled "Identification and role of the key mycotoxin of Pestalotiopsis kenyana, causing leaf spot disease of Zanthoxylum schinifolium " (Manuscript ID: jof-2717955).

Those comments are all valuable and very helpful for revising and improving our paper, as well as the important guiding significance to our research. Revised portions are marked in the paper. The main corrections in the paper are as follows:

To question 1: The section “Introduction”. Lines 46-100: Too wide a range of toxins for a wide range of plant species is presented. 3 toxins are investigated only in this research. Of which only one toxin significantly affected the plant. Therefore, it is advisable to narrow down the spectrum of analyzed toxins in the section “Introduction". Lines 124-141: It would be possible to shorten the overview of information about the role of malondialdehyde.

Answer: We appreciate the reviewer’s important comment. We've removed the broader content and increased the pathogenicity of Pestalotiopsis fungi, “The pathogenic fungus of tea gray blight fungus, P. longiseta, can produce two mycotoxins, oxysporone and ( + ) -epiepoxydon. In addition, P. microspora and P. funereacan can also produce mycotoxins and infect plants.”. In addition, we have also shortened the overview of information on the effects of malondialdehyde. We have deleted the sentence “MDA can react with various cellular components, thereby causing damage to nucleic acids, proteins, and membrane lipids. Thus, MDA is an important indicator of membrane lipid peroxidation. Antioxidant defense enzymes are crucial plant defense systems that scavenge ROS and ensure normal plant growth. In studies conducted on sunflowers infected with S. sclerotiorum, resistant varieties showed significantly higher levels of total protein, carbohydrate, and chlorogenic acid in the leaves than susceptible varieties. This positive correlation between resistance and these indicators was observed, whereas the level of MDA showed a negative correlation. Similar results were observed in chrysanthemum leaves treated with A. alternata toxin solution, with significant increases in soluble protein, MDA, proline content, and PAL, POD, and Polyphenol oxidase (PPO) activities. ”

To question 2: The section “Introduction”. Line 53: Delete repeat, “Leaf Spot of Ixora chinensis, Leaf Spot of Ixora chinensis".

Answer: We appreciate the reviewer’s important comment. We have deleted the redundant “Leaf Spot of Ixora chinensis

To question 3: The section “Introduction”. Line 76: Delete repeat, “by producing toxins, and the toxins".

Answer: We appreciate the reviewer’s important comment. We have deleted the redundant “and the toxins”

To question 4: The section “Introduction”. Line 84: extra dot, “medium. [35,36].”.

Answer: We appreciate the reviewer’s important comment. We have deleted the extra dot.

To question 5: The section “Introduction". Line 100: “pathogenic bacteria”. There may be a typo, because the article is devoted to fungal pathogens.

Answer: We appreciate the reviewer’s important comment. We have changed the “pathogenic bacteria” to “pathogenic fungi”.

To question 6: The section “Introduction”. Lines 142-156: Too blurry a aim with a lot of subtasks. It would be good to state the aim more clearly.

Answer: We appreciate the reviewer’s important comment. We have changed the sentence to “The purpose of this study was to verify whether P. kenyana could produce a mycotoxin to help it cause Z. schinifolium leaf spot and to isolate and purify the mycotoxin. Firstly, the pathogenicity of P. kenyana cultured in different media was determined, and the best medium was selected. Following this, the pathogenic components of P. kenyana mycotoxin were determined to be non-protein through ammonium sulfate precipitation. Preparation of crude mycotoxins, through TLC, silica gel column chromatography, preparation of HPLC separation, and purification of mycotoxins to obtain pure products, and its structure was analyzed. Finally, through the determination of toxicity and physiological and biochemical indicators, it was determined whether mycotoxins are virulence factors that help P. kenyana to induce infection, to provide a theoretical basis for the prevention and control of the leaf spot disease of Z. schinifolium.”

To question 7: The section “Materials and Methods”. Line 158: typo, “Meterials".

Answer: We appreciate the reviewer’s important comment. We have corrected it.

To question 8: The section “Materials and Methods”. Line 165: extra dot, “University. [57]”.

Answer: We appreciate the reviewer’s important comment. We have deleted the extra dot.

To question 9: The section “Materials and Methods”. Line 167: put a dot.

Answer: We appreciate the reviewer’s important comment. We have added a dot.

To question 10: The section “Materials and Methods”. Line 189: “the leaves were placed in Petri dishes”. It is unclear whether the effect of processing in Petri dishes can be extrapolated to live plants.

Answer: We appreciate the reviewer’s important comment. We have changed the sentence to “In vitro infection for leaves was moisturized culture at 25 °C repeated 10 times.” Here is to screen out the best medium for P. kenyana, so the leaves in vitro test were used to detect its toxicity as a reference. For example, the in vitro pathogenicity of the ergot fungus Claviceps purpurea was determined, Inoculation of fresh in vitro-cultivated ovaries with conidial suspension was carried out either by pipetting 0.5 ml droplets onto the stigmas, and it is the same as the plant infection effect. From this, the in vitro pathogenicity assay is an effective alternative to the whole-plant infection tests. https://doi.org/10.1016/j.mycres.2005.11.011

To question 11: The section “Materials and Methods”. Line 207: It is unclear for what reason it is “140 leaves” of Z. schinifolium were taken.

Answer: We appreciate the reviewer’s important comment. We have changed the sentence to “The toxicity test was the same as the previous experiment 2.2.1. Seven saturated precipitation solutions and supernatants in 2.2.2 were applied to the leaves 10 μL, repeated 10 times, and a total of 140 Z. schinifolium leaves were required.”

To question 12: The section “Materials and Methods". Lines 276-277: English translation error, the phrase “Each plant selected 3 branches, each branch selected 5 leaves.”

Answer: We appreciate the reviewer’s important comment. We have changed the sentence to “3 branches per plant, 5 leaves per branch”.

To question 13: The section “Materials and Methods". Line 279: possible typo, the phrase “Each treatment included 10 strains of Z. schinifolium".

Answer: We appreciate the reviewer’s important comment. We have changed the sentence to “Each treatment included 10 plants of Z. schinifolium, 3 branches per plant, 5 leaves per branch, repeated 3 times to observe the incidence.”.

To question 14: The section “Materials and Methods”. Lines 208-209 and 278: there may be a typo in concentration, the phrase “10 µL“ and the phrase "a dose of 1mL per leaf".

Answer: We appreciate the reviewer’s important comment. The previous Lines 208-209 were leaves in vitro test, and 10 μL was added dropwise to each leaf. Line 278 is a potted plant, which was sprayed with 1 mL per leaf.

To question 15: The section “Materials and Methods”. Lines 159-296: a complex scheme of experiments. Perhaps a figure with a diagram of experiments will facilitate the understanding of the study.

Answer: We appreciate the reviewer’s important comment. This is a simple experimental flow figure, and we have put it in the supplementary named Figure S1.

To question 16: The section “Materials and Methods”. Lines 326: you need to specify which significance levels were calculated.

Answer: We appreciate the reviewer’s important comment. We have added “*0.01<p< 0.05; **0.001<p≤0.01; ***p≤0.001.” 

To question 17: The section “Results”. Line 345: “the average of 10 replicates”. Coordinate with the information in the section “Materials and Methods".

Answer: We appreciate the reviewer’s important comment. Here is a writing error, we have changed the material and method 2.2.1 to repeat 10 times.

To question 18: The section “Results”. Lines 356-358: The same effect at all levels of saturation of ammonium sulfate (Table 1). An uninformative table should be moved to the section “Supplementary Materials”.

Answer: We appreciate the reviewer’s important comment. We have moved Table 1. to “Supplementary Material” and changed it to Table S2.

To question 19: The section “Results”. Lines 376-377: probably an error in the number of components (there are not five, but four), the phrase “Component III had very low content, and no clear main point could be determined through TLC. Components I, III, and V,”.

Answer: We appreciate the reviewer’s important comment. Here is a writing error, we have changed V to IV.

To question 20: The section “Results”. Line 385-386: extra dot, “Figure. 4 showed”.

Answer: We appreciate the reviewer’s important comment. Here is a writing error, we have deleted the extra dot.

To question 21: The section “Results”. Line 451 and 457-463 (Figure 8): It is necessary to present the values of the correlation coefficients. It is unclear whether the correlations are positive or negative.

Answer: We appreciate the reviewer’s important comment. We have labeled the value of the correlation coefficient in Figure 8.

To question 22: The section “Results”. Line 449: slang, “time shown”.

Answer: We appreciate the reviewer’s important comment. We have changed the sentence to “time had significant positive correlations with MDA content (p < 0.01)”

To question 23: The section “Conclusions and discussions”. Line 432: it is necessary to explain the reason for the very low chlorophyll content in the variants compared to the “CK1” variant, the differences are 8-16 times. Were there similar intensity effects on other plants?

Answer: We appreciate the reviewer’s important comment. The chlorophyll content of most plants showed a decreasing trend after pathogen infection, but the degree of reduction varied with the type of plant and pathogen. In the discussion, the reason for the extremely low chlorophyll content was explained, and the discussion on the change of chlorophyll content after the application of mycotoxins to other plants was increased.“ When plants are infected by pathogenic fungi, their chlorophyll content often decreases significantly. The total chlorophyll and carotenoid contents of Brassica juncea L.var.Pusa bold seeds treated with 2000 μg / L aflatoxin B1 decreased by 67.61 % and 82.70 %, respectively. After the treatment of corn with the toxic Compounds of Exserohilum turcicum, reduction in total chlorophyll, chlorophyll a, and chlorophyll b content was 49.4%, 38.6%, and 64.1%.” Our experiment showed a significant decrease in chlorophyll content in Z. schinifolium leaves with treatment time, suggesting a destructive effect on chloroplasts or inhibitory effects on chlorophyll synthesis by the mycotoxin. ”

To question 24: The section “Conclusions and discussions”. Lines 490-491: incorrect grouping by category, “including peptides, proteins, sugars, lipids, aromatic rings, heterocycles, and organic acids.”

Answer: We appreciate the reviewer’s important comment. We have changed the sentence to “Mycotoxins belong to various chemical categories, including proteins, esters, ketones, sugars, organic acids and alkaloids.”.

To question 25: The section “References". Check the references. Lines 716-718: for example, the bibliographic reference [58] is incomplete.

Answer: We appreciate the reviewer’s important comment. We have changed the reference [58] to “Tang, X.; Zhang, J.Q.; Jiang, W.K.; Yuan, Q.S.; Wang, Y.H.; Guo, L. P.; Yang, Y. Yang, Y.; Zhou, T. Isolation, identification, and pathogenicity research of brown rot pathogens from Gastrodia elata. China Journal of Chinese Materia Medica. 2022, 47, 2288-2295. https://doi.org/10.19540/j.cnki.cjcmm.20220223.102”, and we have checked all the references.

We tried our best to improve the manuscript and made every change in the manuscript according to the reviewers’ comments. These changes will not influence the content and framework of the paper.

We appreciate for editor and reviewers’ warm work earnestly, and hope that the correction will meet with approval.

Once again, thank you very much for your comments and suggestions.

Yours sincerely,

Chang Liu, Yiling Li

Round 2

Reviewer 1 Report

Comments and Suggestions for Authors

Still needs major revision.

See my comments in the attached pdf file

Comments on the Quality of English Language

The same like the first version

Author Response

Modification instructions

Dear Reviewer,

    Thank you for your comments concerning our manuscript entitled "Identification and role of the key mycotoxin of Pestalotiopsis kenyana, causing leaf spot disease of Zanthoxylum schinifolium " (Manuscript ID: jof-2717955).

    Those comments are all valuable and very helpful for revising and improving our paper, as well as the important guiding significance to our research. Revised portions are marked in the paper. The main corrections in the paper are as follows:

To question 1: P2L20-22 Improve the language.

Answer: We appreciate the reviewer’s important comment. We have changed the sentence to “PK-3 had the highest toxicity to Z. schinifolium, which may be the primary mycotoxin, according to the biological activity test by spray method.”.

To question 2: P2L23 treated with

Answer: We appreciate the reviewer’s important comment. We have changed the sentence to “The physiological and biochemical indexes of Z. schinifolium plants treated with PK-3 mycotoxin were determined.”

To question 3: P2L24-25 improve the language again

Answer: We appreciate the reviewer’s important comment. We have changed the sentence to “The results within 35 days after mycotoxin treatment showed that the protein content and malondialdehyde content of leaves increased over time.”

To question 4: P2L26 over time

Answer: We appreciate the reviewer’s important comment. We have changed the sentence to “The soluble sugar content and chlorophyll content decreased over time.”

To question 5: P2L31-34 reformulate again, and remove the last part of the sentence.

Answer: We appreciate the reviewer’s important comment. We have changed the sentence to “In this study, the pathogenic mechanism of Z. schinifolium leaf spot was discussed, offering a theoretical foundation for improved disease prevention and control”, and we have removed the last part of the sentence.

To question 6: P3L61,81-82 improve the language. The disease is not caused by producing mycotoxins.

Answer: We appreciate the reviewer’s important comment. We have changed the sentence to “Pestalotiopsis can produce abundant metabolites and can also produce mycotoxins as virulence factors to induce pathogenic fungi to infect host plants. The reported mycotoxins are often ketones.”.

To question 7: P3L91-92 reformulate again

Answer: We appreciate the reviewer’s important comment. We have changed the sentence to “During the culture, it is common for pathogenic fungi to secrete their mycotoxins into the medium [24,25].”.

To question 8: P3L108 fungus (fungi is plural)

Answer: We appreciate the reviewer’s important comment. We are very sorry for our incorrect writing and have changed the fungi to fungus.

To question 9: P5L207-208 reformulate

Answer: We appreciate the reviewer’s important comment. We have changed the sentence to “Leaves in vitro infections were moisturized at 25℃ and repeated 10 times [41].”.

To question 10: P6L216 fungal secondary metabolites

Answer: We appreciate the reviewer’s important comment. We have changed the sentence to “2.2.2. Determination of toxic components of fungal secondary metabolites”.

To question 11: P6L219 fungal mycelium

Answer: We appreciate the reviewer’s important comment. We have changed the fungi to fungal mycelium.

To question 12: P7L278 UV scanning

Answer: We appreciate the reviewer’s important comment. We have changed the sentence to “UV-visible spectrophotometer scanning” and we also modified in 3.2.2. We have changed the sentence to “3.2.2. UV-visible spectrophotometer scanning; PK-1, PK-2, and PK-3 were scanned using a UV-visible spectrophotometer, and methanol as the control; Figure 4. UV-visible spectrophotometer scanning diagram.” 

To question 13: P7L310 treatment

Answer: We appreciate the reviewer’s important comment. We have changed the infection to treatment.

To question 14: P9L357-358 toxic components of fungal secondary metabolites

Answer: We appreciate the reviewer’s important comment. We have changed the sentence to “Screening of the best medium for P. kenyana to produce mycotoxin and determination of the toxin components of fungal secondary metabolites”.

To question 15: P9L359 please understand and distinguish between the fungus itself and its metabolites. In this case you are talking about the treatment of the leaves with the media filtrates that contain mycotoxins. i.e. 2.2.1. Preparation of mycotoxin stock solution and determination of toxicity. This means that these lesions are induced by these metabolites not by the fungus itself.

Answer: We appreciate the reviewer’s important comment. We have changed the sentence to “The toxicity test revealed the metabolites of P. kenyana cultured in four different media induced yellow-brown lesions on the leaves of Z. schinifolium, but their toxicity varied ”.

To question 16: P9L361, 363 the metabolites of the fungus (fungi is plural) cultures in... 

Answer: We appreciate the reviewer’s important comment. We have changed the sentence to “Among the four media, the metabolites of the fungus cultures in PDB resulted in the highest lesion area, accounting for 57.17% of the total leaf area, signifying the strongest pathogenicity. The metabolites of the fungus cultures in modified martin broth, potato sucrose broth, and czapek dox broth exhibited lesser toxicity, with lesion areas accounting for 23.10%, 20.68%, and 19.38% of the total leaf area, respectively. ”.

To question 17: P9L370 The effect of metabolites of......

Answer: We appreciate the reviewer’s important comment. We have changed the sentence to “Figure 1. The effect of metabolites of P. kenyana cultivated in different media on the lesion area of Z. schinifolium.”

To question 18: P9L372 please correct it in Fig. 1.F (in the graph) Or introduce your abbreviation. What does CK mean?

Answer: We appreciate the reviewer’s important comment. We have corrected it in Fig. 1.

To question 19: P9L373-376 toxicity of metabolites of; it is not a disease. represent the lesion caused by the fungal metabolites of P. kenyana; metabolites of P. kenyana

Answer: We appreciate the reviewer’s important comment. We have changed the sentence to “A-E show the toxicity of metabolites of P. kenyana cultured in different media to Z. schinifolium. The yellow-brown spots in the image represent the lesion caused by the fungal metabolites of P. kenyana. The data in F represents the percentage of lesion area to total leaf area (%) and is the average of 10 replicates. Different lowercase letters indicate significant differences in the lesion area on Z. schinifolium leaves caused by metabolites of P. kenyana cultivated on different media at a significance level of p < 0.05 (LSD method).”

To question 20: P15L470 Please enhance the resolution of the graph. The resolution of the former graph was much better.

Answer: We appreciate the reviewer’s important comment. We have replaced the higher resolution figure.

To question 21: P15L470 please correct it in Fig. 7. (in the legend). Or introduce your abbreviation. What does CK mean?

Answer: We appreciate the reviewer’s important comment. We have corrected it in Fig. 7.

To question 22: P16L479-480 toxicity; to z. schinifolium are similar.

Answer: We appreciate the reviewer’s important comment. We have changed the sentence to “The results showed that PK-3 toxicity and P. kenyana pathogenicity to Z. schinifolium are similar.”

To question 23: P17L499 see my comment in the first version. what do you mean with incidence.

Answer: We appreciate the reviewer’s important comment. Incidence means the percentage of lesion area to total leaf area of Z. schinifolium leaves treated with PK-3 mycotoxins. We have changed the sentence to “Figure 8. The percentage of lesion area to total leaf area and correlation analysis of Z. schinifolium treated with PK-3 mycotoxin.”

To question 24: P17L500-502 reformulate again

Answer: We appreciate the reviewer’s important comment. We have changed the sentence to “Note: A, The percentage of lesion area to total leaf area of Z. schinifolium leaves treated with PK-3 mycotoxins; B, The correlation between the proportion of lesions and time, mycotoxin concentration and physiological and biochemical indexes of Z. schinifolium after PK-3 mycotoxin treatment.”

To question 25: P17L502 the same comment on CK abbreviation

Answer: We appreciate the reviewer’s important comment. We have corrected it in Fig. 8.

To question 26:P17L531-533 the metabolites of....

Answer: We appreciate the reviewer’s important comment. We have changed the sentence to “According to the results of the toxicity test, the metabolites of P. kenyana cultured in PDB in this experiment caused the largest lesion area of Z. schinifolium leaves, that is, the strongest toxicity. Therefore, among the four tested media, the metabolites of P. kenyana cultured in PDB had the strongest toxicity and were used for subsequent culture.”

To question 27:P18L536-538 reformulate the entire paragraph from line 536 to 569.

Please show what other authors have done and compare your results with their results as you did in the previous paragraph. Use ONLY references that are relevant to the topic and compare with fungi from the same genus. And please shorten the paragraph.

Answer: We appreciate the reviewer’s important comment. We have changed the paragraph to “Mycotoxins belong to various chemical categories, including proteins, esters, ketones, sugars, organic acids, and alkaloids. Before separating and purifying mycotoxins, it is necessary to determine whether mycotoxins are proteins, and different separation methods are adopted accordingly. Therefore, in this experiment, the mycotoxins of P. kenyana were first determined to be non-protein substances. Organic solvents are often used to extract and enrich the fermentation broth, which is then detected and separated by chromatographic techniques. P. sydowiana solid culture was extracted with ethyl acetate, and 10 compounds were isolated by reversed-phase high performance liquid chromatography [55]. P. karstenii fermentation broth was subjected to silica gel CC using petroleum ether/ethyl acetate, and four compounds were isolated [56]. In this experiment, ethyl acetate was used for multiple extraction and enrichment to obtain crude mycotoxin. The crude toxin of P. kenyana was subjected to silica gel CC and HPLC purification using petroleum ether: ethyl acetate = 3: 1 system, and a total of 3 compounds were obtained. Few compounds were isolated, which may be due to the limited sample size, some low-content components not being separated, or improper operation in the process of crude toxin preparation and chromatography, which resulted in a poor separation effect. In addition, due to the complexity of mycotoxin components, a combination of multiple developing agent systems is sometimes required. The structure of the mycotoxins obtained by separation needs to be analyzed by NMR, MS, and other means. For example, the compounds isolated from the culture of P. theae were identified as Pestalazines A and B, and Pestalamides A-C by nuclear magnetic resonance spectroscopy [57]. The structures of the three compounds were examined using nuclear magnetic resonance hydrogen and carbon spectra in this experiment.”

To question 28:P18L570-576this is just literature review not discussion. please compare with your results.

Answer: We appreciate the reviewer’s important comment. We have increased the comparison with the results of this experiment and changed the paragraph to “Among them, PK-3, Pestalopyrone, has been reported as a fungal metabolite and mycotoxin. Sterigmatocystin and Pestalopyrone were isolated from the extracts of the mangrove endophytic fungus Nigrospora oryzae [58]. Pestalopyrone, a mycotoxin produced by P. oenotherae, is 6-(1′-methylprop-1′-enyl)-4-methoxy-2-pyrone. Pestalopyrone was also isolated from P. guepinii culture filtrate [22]. Compared with the previous studies on the pathogenic fungi of Pestalotiopsis sp., the same metabolites were isolated from P. kenyana in this experiment, which indicated that Pestalopyrone may be a virulence factor produced by the pathogenic fungi of Pestalotiopsis sp., inducing pathogenic fungi to cause pathogenic effects on its host plants. Therefore, it is speculated that Pestalopyrone is a toxic substance produced by P. kenyana

To question 29:P19L583 treatment

Answer: We appreciate the reviewer’s important comment. We have changed the infection to treatment.

To question 30:P19L585 Start a new paragraph here.

Answer: We appreciate the reviewer’s important comment. We have started a new paragraph here.

To question 31:P20L629-638 This is not a conclusion. this is just summery for your results.

Answer: We appreciate the reviewer’s important comment. We have changed the paragraph to “In summary, this study found that PDB was the best medium for P. kenyana to produce mycotoxins, and PK-3 was a non-protein mycotoxin with strong toxicity to Z. schinifolium. It suggested Pestalopyrone was a virulence factor that induced the pathogenicity of P. kenyana to Z. schinifolium. The current study advances our understanding of Pestalopyrone and further studies on the pathogenic mechanism of Z. schinifolium leaf spot caused by P. kenyana.”. 
